# Field Evaluation of Experimental Maize Hybrids for Resistance to the Fall Armyworm (Lepidoptera: Noctuidae) in a Warm Temperate Climate

**DOI:** 10.3390/insects15040289

**Published:** 2024-04-19

**Authors:** Xinzhi Ni, Alisa Huffaker, Eric A. Schmelz, Wenwei Xu, W. Paul Williams, Baozhu Guo, Xianchun Li, Fangneng Huang

**Affiliations:** 1United States Department of Agriculture-Agricultural Research Service, Crop Genetics and Breeding Research Unit, Tifton, GA 31793, USA; baozhu.guo@usda.gov; 2Division of Biological Science, University of California-San Diego, La Jolla, CA 92093, USA; ahuffaker@ucsd.edu (A.H.); eschmelz@ucsd.edu (E.A.S.); 3Agricultural Research & Extension Center, Texas A&M AgriLife Research, Texas A&M University System, Lubbock, TX 79403, USA; wxu@ag.tamu.edu; 4United States Department of Agriculture-Agricultural Research Service, Corn Host Plant Resistance Research Unit, Mississippi State, MS 39762, USA; pwilliams0961@gmail.com; 5Department of Entomology, University of Arizona, Tucson, AZ 85721, USA; lxc@arizona.edu; 6Department of Entomology, Louisiana State University Agricultural Center, Baton Rouge, LA 70803, USA; fhuang@agcenter.lsu.edu

**Keywords:** fall armyworm resistance, maternal/cytoplasmic effect, predators, phytoalexins, principal component analysis

## Abstract

**Simple Summary:**

To develop new maize germplasm, and understand the genetic basis of fall armyworm resistance, 12 experimental hybrids (six sets of reciprocal crosses) with diverse genetic backgrounds were compared with four commercial checks. Reciprocal crosses (an inbred line was used in a pair of crosses as male and female parents) were used to determine the maternal effect of an inbred line on fall armyworm resistance. Fall armyworm resistance was assessed using its injury ratings on artificially infested maize plants, and possibly plant attraction to predators of the fall armyworm larvae. Two reciprocal crosses (‘FAW1430’ × ‘Oh43’ and ‘CML333’ × ‘NC358’) showed the least fall armyworm injury. A total of 11 taxa of predators (five lady beetles and five other insect predators, and spiders) were recorded. However, the number of predators was not negatively correlated to fall armyworm injury. The predator data also varied greatly between the two years when the experiment was conducted. In addition, both parents contributed similarly and no maternal effect on fall armyworm resistance was detected.

**Abstract:**

The polyphagous fall armyworm (FAW), *Spodoptera frugiperda*, has become an invasive pest worldwide in recent years. To develop maize germplasm with multiple pest resistance and understand genetic inheritance, 12 experimental hybrids (six pairs of reciprocal crosses) with diverse genetic backgrounds and four commercial checks were examined for FAW resistance in 2013 and 2014. The experiment utilized a randomized complete block design with four replications as the block factor. FAW injury on maize plants was assessed at 7 and 14 d after the artificial infestation at the V6 stage, and predatory arthropod taxa and abundance on maize seedlings were recorded 7 d after the infestation. *Spodoptera frugiperda* resistance varied significantly among the 16 hybrids. Two reciprocal crosses (‘FAW1430’ × ‘Oh43’ and ‘CML333’ × ‘NC358’) showed the least FAW injury. Eleven arthropod predators [i.e., six coleopterans, three hemipterans, earwigs (dermapterans), and spiders (or arachnids)] were also recorded; the two most common predators were the pink spotted ladybeetle, *Coleomegilla maculata*, and the insidious flower (or minute pirate) bug, *Orius* spp. Predator abundance was not correlated to FAW injury but varied greatly between 2013 and 2014. Principal component analysis demonstrated that, when compared with FAW resistant (or Bt-transgenic) checks (‘DKC69-71’, ‘DKC67-88’, and ‘P31P42’), five pairs of the reciprocal crosses had moderate FAW resistance, whereas a pair of reciprocal crosses (‘NC350’ × ‘NC358’ and NC358 × NC350) showed the same FAW susceptibility as the non-Bt susceptible check ‘DKC69-72’. Both parents contributed similarly to FAW resistance, or no maternal/cytoplasmic effect was detected in the experimental hybrids.

## 1. Introduction

Biotic and abiotic stresses are the most important impediments to maize production in a warm temperate climate in the southern U.S. states. With exacerbation of the ongoing trend of climate change, the Midwest Corn Belt states are projected to become unsuitable for maize production by 2100 [1]. Maize germplasm that confers resistance to multiple biotic and abiotic stresses are of critical importance. Maize germplasm resistance to individual biotic (e.g., insect and disease) or abiotic (e.g., drought and heat) stress factors has been reported repeatedly, but only a relatively limited number of reports on developing germplasm lines resistant to multiple stress factors are available. Although studies have been conducted in recent decades to examine native resistance to multiple insect species in maize germplasm and hybrids [2,3,4,5,6], few studies have simultaneously examined resistance to both insects and diseases, and reduced mycotoxin contaminations. Of all biotic stresses in a warm temperate/sub-tropical climate, e.g., Tifton, GA (with GPS coordinates of 31.5° N and 83.5° W), in the southern USA, the fall armyworm (FAW) [*Spodoptera frugiperda* (J.E. Smith) (Lepidoptera: Noctuidae)] is an important foliar-feeding insect, and the corn earworm [*Helicoverpa zea* (Boddie) (Lepidoptera: Noctuidae)] and occasional *S. frugiperda* infestations are the key ear-feeding insects at the post-flowering stage. In particular, with climate change, development of maize germplasm lines with resistance to multiple biotic and abiotic stress factors and reduced aflatoxin contamination is not only critically important now to maize production in the southern U.S. states, but also significantly beneficial to future maize production in the northern states, e.g., the Midwest Corn Belt. This is because the biotic and abiotic stress factors currently present in the warm temperate/subtropical climate would likely be encountered by growers in the U.S. Midwest Corn Belt, and other maize production regions worldwide, in less than eight decades, or by 2100 [1].

One of the long-term goals for our research program has been to develop new maize germplasm with resistance to multiple biotic and abiotic stresses, reduced aflatoxin contamination, and good agronomic traits (e.g., yield potential). In addition, parental (maternal/paternal) contributions to insect resistance traits in a hybrid from each parental inbred line in maize could be critical, because resistance to different insect pests in maize can be inherited in different ways based on the literature related to host plant resistance to insects. Disease and insect susceptibility/resistance in crop plants varies with male or female parent, which is also used to assess maternal and cytoplasmic effect. The phenomenon of maternal and cytoplasmic effect has been reported in disease resistance in maize [7], rice [8], durum wheat [9], and olive [10]. Resistance to the southern corn leaf blight (*Cochliobolus heterostrophus*) and the yellow corn leaf blight (*Mycosphaerella zeae-maydis*) in maize was associated with the maternally inherited T (Texas source) of male-sterile cytoplasm [7]. However, maize resistance to the northern corn leaf blight, *Helminthosporium turcicum,* showed no maternal or cytoplasmic effect [11]. Similar phenomena were observed in insect resistance in maize. Pericarp of maize kernels did show a maternal effect in maize weevil resistance [12], but maysin and its analogs in maize that contribute to *H. zea* resistance did not show a maternal effect [13]. In addition, previous reports suggest a correlation among genes that confer resistance to multiple insects on maize, e.g., genes conferring resistance to foliar-feeding *S. frugiperda* and the southwestern corn borer, *Diatraea grandiosella* Dyar (Lepidoptera: Crambidae), and genes conferring resistance to ear-feeding European corn borer, *Ostrinia nubilalis* (Hübner) (Lepidoptera: Crambidae), and *H. zea* [2,14,15,16]. Also, tropical maize inbred lines ‘CML71’, ‘CML125’, ‘CML370’ and ‘CKSBL10008’, derived from multiple borer resistant inbred lines, demonstrated that CKSBL10008 and CML71 confer high levels of antibiosis to *S. frugiperda* [17].

Because our approach for new germplasm development has been to utilize both basic and applied research findings on maize germplasm improvement with resistance to multiple stress factors, the reciprocal crosses (as shown in Table 1) were designed and made in 2012 to develop germplasm lines with not only multiple pest resistance, but also to determine maternal and/or cytoplasmic contributions to multiple insect and disease resistance in these experimental hybrids. The experimental hybrids used eight inbred lines and two newly available maize inbred lines with tropical backgrounds (i.e., ‘CML528’ from CIMMYT, and FAW1430 from our program), and six parental inbred lines of the nested association mapping (NAM) populations (or NAM founders). These six NAM founders had high levels of phytoalexins, i.e., kauralexins [18] and zealexins [19]. At the same time, predator survey data were included for assessing negative correlations between the number of predators and plant injury ratings, which could possibly be used as an additional parameter for comprehensively assessing host plant resistance to insects [6,20]. However, the previous findings showed that the results varied greatly from year to year.

Using the 12 experimental hybrids (i.e., six sets of reciprocal crosses) and the four commercial hybrids as the checks (or controls), the four objectives of the current two-year study were to (1) determine *S. frugiperda* resistance and predator abundance among the hybrids; (2) determine the maternal effect on *S. frugiperda* resistance and predator abundance; (3) examine the correlation between *S. frugiperda* injury rating and predator abundance; and (4) identify the best experimental hybrids to be used for new germplasm development. While the results of the evaluation of *S. frugiperda* resistance in 2013 and 2014 are presented herein, the results of the evaluations of ear- and kernel-feeding insect and disease resistance, and yield, as well as the aflatoxin accumulation levels of these hybrids, will be reported separately in two other manuscripts (Ni et al., in review).

## 2. Materials and Methods

### 2.1. Parental Selection for the Breeding Crosses

A total of 12 (or six sets) of reciprocal breeding crosses were made in 2012 utilizing eight inbred lines with unique genetic backgrounds, as shown in Table 1. Of all parental inbred lines, ‘FAW1430’, an *S. frugiperda*-resistant inbred line (to be released) from our laboratory, is derived from a maize population ‘GT-FAWCC(C5)’, which had been released [22] and subsequently examined [25,26]. ‘CML528’ is a new tropical germplasm line selected from a set of 12 new releases (‘CML524’ − ‘CML535’) that our program procured in 2011 from the CIMMYT (Mexico City, Mexico).

The remaining inbred lines are the six NAM founders with relatively high levels of phytoalexins, i.e., kauralexins [18] and zealexins [19]. The four NAM founders (i.e., ‘B97’, ‘CML333’, ‘NC350’, and ‘NC358’) have relatively high levels of both kauralexins and zealexins. The fifth NAM founder ‘Oh43’ has a relatively high level of kauralexins, but a low level of zealexins [18,19], and Oh43 is also known to be *S. frugiperda* resistant with a high level of a trypsin inhibitor, or Bowman–Birk inhibitor [21]. The last NAM founder, ‘CML322’ with white kernels (for food corn), was also reported to have relatively low levels of both kauralexins and zealexins [18,19]. In addition, CML322 not only produced grains with consistently low levels of aflatoxin contamination [23], but also showed drought tolerance [27].

In addition, four commercial maize hybrids used as the checks (CKs) for the experiment were ‘DKC69-71’, ‘DKC69-72’, ‘DKC67-88’, and ‘P31P42’. DKC69-71 is a hybrid of the YieldGard Corn Borer^®^ trait with transgenes of *Bacillus thuringiensis* (Bt), Cry1Ab, for lepidopteran pest control and Roundup Ready 2 (RR2) for herbicide resistance, while DKC69-72 is an isogenic non-Bt hybrid of DKC69-71. DKC67-88 is a hybrid of the Genuity^®^ VT Triple PRO^®^ traits containing two Bt transgenes (Cry1A.105/Cry2Ab2) for managing lepidopteran pests, one Bt gene (Cry3Bb1) for controlling underground coleopteran rootworms, and the RR2 gene for herbicide resistance. P31P42 is a hybrid of Herculex I trait possessing Cry1F Bt gene for controlling lepidopteran pests, and two herbicide resistance genes RR2 and LibertyLink^®^ (LL). The first three hybrids were provided from Bayer Crop Science (formerly Monsanto, St. Louis, MO, USA), while P31P42 was from Corteva Agriscience (formerly Pioneer Hi-Bred International, Inc., Johnston, IA, USA).

### 2.2. Sources of Insects

The neonate larvae of *S. frugiperda* used in this study (in both 2013 and 2014) were from two laboratory colonies maintained at the Insectaries of the Corn Host Plant Resistance Research Unit, USDA-ARS, at Mississippi State, MS, USA and the former Plant Protection and Management Research Unit, USDA-ARS, at Tifton, GA. The *S. frugiperda* colony was established originally using insects collected from maize plants, and feral insects were collected and added annually to the colony to maintain insect colony performance. The larvae of *S. frugiperda* used in this study were considered as maize strain. All predators recorded in the experimental plots were attracted naturally by the maize seedlings at stages V6 (at the infestation) and V7 (7 d after the infestation) in the springs of 2013 and 2014, respectively.

### 2.3. Artificial S. frugiperda Infestation and Injury Rating at the Seedling (Whorl) Stage

The *S. frugiperda* infestation was established by manually infesting the whorl leaf tissue of all maize plants in an experimental plot at 6-leaf (V6) stage using a mixture of newly hatched neonate larvae and corn cob grits (F and S Equipment and Supplies Inc., Birmingham, AL, USA). The artificial infestation dates were 12 May 2013, and 20 May 2014. A modified procedure as previously described [28,29] was used for the artificial infestation of all plants in each experimental plot. Briefly, 25–35 neonate larvae per plant were used with Bazooka insect infestation apparatus. The number of insects (25–35) per 0.2 mL drop of the mixture was produced by a mixing ratio of 1 mL neonate larvae with 30 mL corn cob grits. The levels of insect injury were then rated at 7 and 14 d after the infestation using a modified visual rating scale of 1–9, as described previously [6,28,30]. The modified scale used for the *S. frugiperda* injury ratings is as follows: 1 = No injury or few pin-holes; 2 = Few short holes (also known as shot holes) on several leaves; 3 = Short holes or lesion up to 1.3 cm on several leaves; 4 = Loss of whorl, only top leaves with most short holes and a few long lesions (≤2.5 cm); 5 = Loss of whorl, only top leaves with more short than long lesions; 6 = Loss of whorl, only top leaves with long lesions; 7 = Loss of whorl, long lesions common on one half of the leaves, loss of portion of top leaves; 8 = Loss of whorl, long lesions common on one half to two thirds of leaves, loss of top leaves; 9 = Loss of whorl, most leaves with long lesions, and complete defoliation was observed. The visual insect injury rating was an overall rating of *S. frugiperda* injury on all plants in an experimental plot under the field conditions.

### 2.4. Predator Abundance Survey at the Seedling Stage

Both the taxa and abundance of predatory arthropods from each experimental plot were recorded 7 d after the initial *S. frugiperda* infestation, as described previously [6,20]. Data were collected (on 19 May 2013, and 27 May 2014) by careful visual observation of predators on all plants per single-row plot with minimal disturbance to predator activities. All recorded predators were identified according to their genus or species, except earwigs and spiders, which were also recorded as a single taxon, and no further taxonomic identification was performed in the current study. Total numbers of predators were then calculated. In addition, the numbers of predator taxa recorded on each hybrid entry were also compared to understand the predator diversity among the breeding crosses used in the experiment. Although parasitoids are often associated with *S. frugiperda* infestations in maize fields [31,32,33], few parasitoids were observed 7 d after the artificial infestations with neonate larvae of *S. frugiperda* in the spring. Thus, the parasitoids were not included in the sampling.

### 2.5. Study Site, Experimental Design, and Data Analysis

The experiments were conducted in 2013 and 2014 on the Belflower Research Farm near Tifton, GA, USA (31.5° N, 83.5° W), which is considered to have a warm temperate/subtropical climate. Weather data used for the data analysis were obtained from the Georgia Weather website [http://www.georgiaweather.net/ (accessed on 14 March 2024)]. The same trials were planted on 11 April 2013, and 4 April 2014 for field evaluation. The experiments in both 2013 and 2014 utilized a randomized complete block design with four replications as the block factor. Within each experimental block (or replication), there were randomized single-row plots (approximately 5m × 1m) for each experimental hybrid entry. Due to the limited seed supplies of the 12 selected experimental hybrids, which were originated from a diallel cross plan, a single-row plot was used in the current study. Each experiment conducted in 2013 and 2014 was considered a separate trial.

The *S. frugiperda* feeding injury ratings and predators recorded on the maize leaves were analyzed using analysis of variance (PROC MIXED procedure) followed by the Fisher’s Protected LSD test (α = 0.05) using statistical analysis software (SAS 9.4, 2016, SAS Institute, Cary, NC, USA). In the PROC MIXED model, the breeding cross entry and year were considered the fixed factors to assess both inherited resistance and genotype by environment (G × E) interactions, whereas four replications within each year were considered the random factor. The predator data were log-transformed because of the high coefficient of variation (or the ration of the standard deviation to the mean) of the predator dataset as shown in Appendix A. To further decipher G × E interactions, correlations between predator profiles and *S. frugiperda* injury ratings at the seedling stage were analyzed to understand the relationships among all 15 parameters assessed in the study (as shown in Appendix A). The relationships were assessed using Pearson’s correlation coefficient from the PROC CORR statement of the SAS software. Weather data (i.e., daily high and low temperatures and rainfall) from May in 2013 and 2014 were compared (using paired *t*-tests) to understand environmental influences on *S. frugiperda* injury ratings and predator abundance.

To identify the best hybrids with resistance to both *S. frugiperda* and possible predator attraction at the seedling stage of maize plants, the overall (two-year) means of *S. frugiperda* injury ratings and log-transformed predator data were subjected to the principal component analysis (PROC PRINCOMP) procedure. To ensure the data were in the same direction for the PROC PRINCOMP, procedure as described in [34], the predator abundance data were transformed by “×(−1)” to convert the predator abundance data to negative values, so that predator data were in the same direction as the insect injury rating data.

Furthermore, because the cytoplasmic/maternal effect is important in understanding parental contributions to pest resistance and, subsequently, the targeted design of breeding crosses and germplasm development, the cytoplasmic/maternal effect on *S. frugiperda* resistance in the six sets of reciprocal crosses was further examined using paired *t*-tests (α = 0.05) with the SAS software (version 9.4).

## 3. Results

### 3.1. The S. frugiperda Injury at the Seedling Stage

The pooled *S. frugiperda* injury ratings were significantly different among the hybrids (*F* = 18.63; df = 15, 220; *p* < 0.0001), and between the sampling dates (7- and 14-d) (*F* = 61.82; df = 1, 220; *p* < 0.0001), and so was hybrid by sampling day (7 or 14 d) interaction (*F* = 2.22; df = 15, 220; *p* = 0.007). However, *S. frugiperda* injury ratings were not different between 2013 and 2014 (*F* = 1.32; df = 1, 220; *p* = 0.25). Thus, the data from both years were combined for further analysis.

The combined *S. frugiperda* injury ratings in 2013 and 2014 showed that *S. frugiperda* injuries were significantly different among the 16 entries after 7 d (Figure 1A) and 14 d (Figure 1B), as well as for the overall means (Figure 1C). All 12 breeding crosses showed moderate resistance to *S. frugiperda*, when compared to the susceptible check DKC69-72 and the three transgenic *Bt* hybrids (DKC69-71, DKC67-88 and P31P42) with *Bt* toxins. Of the 12 experimental hybrids, FAW1430 × Oh43 and CML333 × NC358 were the two best experimental hybrids (indicated by the asterisks on the bars in Figure 1C), because their *S. frugiperda* injury ratings were below the mean (2.24 ± 0.06, *n* = 256) of all 16 hybrid entries (indicated by the horizontal line across the bar graph). At the same time, the *S. frugiperda* injury ratings from neither set of the reciprocal crosses differed.

In addition, a strong influence of hybrid by year interaction on *S. frugiperda* injuries at both 7 d (*F* = 5.19; df = 15, 90; *p* < 0.0001) and 14 d (*F* = 2.69; df = 15, 90; *p* = 0.002) was detected. Because hybrid by sampling day interaction influenced the injury ratings significantly, the 7 and 14 d injury ratings and means from 2013 and 2014 are presented separately, as shown in the Appendix A (Appendix A for 2013 data; and Appendix A for 2014 data). The injury ratings also differed between the two years on the 7 d injury ratings (*F* = 12.79; df = 1, 6; *p* = 0.01). In 2013, the 7 d injury rating was 1.81 ± 0.08 (*n* = 64) (Appendix A), but in 2014, it was 2.02 ± 0.07 (*n* = 64) (Appendix A). However, there was no difference in the 14 d injury rating between 2013 and 2014 (*F* = 0.03; df = 1, 6; *p* = 0.87), which are shown in Appendix A (14 d rating in 2013—2.57 ± 0.14, *n* = 64) and Appendix A (14 d rating in 2014—2.55 ± 0.16, *n* = 64), respectively. Thus, contribution of predator types and abundance on *S. frugiperda* injury was further evaluated.

### 3.2. Predator Abundance and Diversity Survey

A total of 11 arthropod predators [i.e., six coleopterans, three hemipterans, earwigs (dermapterans), and spiders (or arachnids)] were recorded on maize plants (at the V7 stage) 7 d after the artificial *S. frugiperda* infestation. The results were similar to the previous reports [6,20]. The five lady beetles were as follows: the convergent lady beetle, *Hippodamia convergens* (Coleoptera: Coccinellidae), the pink-spotted lady beetle, *Coleomegilla maculata* (Coleoptera: Coccinellidae); the seven-spotted lady beetle, *Coccinella septempunctata* (Coleoptera: Coccinellidae), the multicolored Asian lady beetle, *Harmonia axyridis* (Coleoptera: Coccinellidae), and the dusky lady beetle, *Scymnus* spp. (Coleoptera: Coccinellidae). The hooded beetle is also known as the flower beetle, *Notoxus* spp. (Coleoptera: Anthicidae). The three hemipteran predators were the big-eyed bug, *Geocoris* spp. (Hemiptera: Geocoridae), the insidious flower bug (or minute pirate bug), *Orius insidiosus* (Hemiptera: Anthocoridae), and the damsel bug, *Nabis* spp. (Hemiptera: Nabidae). The earwigs were identified as *Labidura riparia* (Labiduridae), and *Doru taeniatum* (Forficulidae). All spider species were recoded as one group, and no further identification was made. The pooled data showed that the total number of predators did not vary significantly among the hybrid entries (*F* = 0.63; df = 15, 90; *p* = 0.84). Because there were no significant differences among the hybrid entries, except the difference between the two years, the pooled two-year data were not presented.

However, the predators varied significantly between the two years (*F* = 12.95; df = 1, 6; *p* = 0.01), although hybrid entry by year interaction did not affect predator abundance (*F* = 0.71; df = 15, 90; *p* = 0.77). Although the two most common predators observed on experimental plots in both years were the pink spotted lady beetle and the insidious flower bug, the abundance of all predators recorded in 2013 was only half of that in 2014, as shown in the Appendix A (Appendix A). The predators per plot were nearly doubled in 2014 (4.42 ± 0.31, *n* = 64) when compared to the data from 2013 (2.22 ± 0.17, *n* = 64) (Appendix A).

Although the 7 d injury ratings in 2013 (Appendix A) was less than in 2014 (Appendix A), the 14-d injury ratings were not different between 2013 (Appendix A) and 2014 (Appendix A). The nearly doubled predator populations recorded at 7 d after the artificial infestation in 2014, when compared to the finding in 2013, suggested that the increased predation resulted in reduced *S. frugiperda* survival, which might have resulted in no significant difference in 14 d injury ratings between 2013 and 2014, as described previously.

Comparison of weather data for May between 2013 and 2014 showed that the daily high temperature was greater in 2014 (28.66 ± 0.65, *n* = 31) than in 2013 (26.66 ± 0.70, *n* = 31), and the difference was 2.01 ± 0.8 °C (*n* = 31, paired t-test *t* = 2.5, df = 30, *p* = 0.02). However, neither daily low temperature nor rainfall differed (*p* values > 0.13). Thus, lower daily high temperature in May of 2013 than that in 2014 might have caused less feeding by early instar larvae and subsequently lower *S. frugiperda* injury ratings at day 7 in 2013 (Appendix A) than in 2014 (Appendix A). In contrast, the nearly doubled number of predators recorded on day 7 after the artificial infestation in 2014 than in 2013 (Appendix A), caused by relatively higher daily high temperature, might have contributed to the reduced *S. frugiperda* survival, due to predation, and subsequently no significant difference in the 14 d injury ratings between the two years.

### 3.3. Correlation between S. frugiperda Injury Ratings and among the Predators

Based on the combined two-year data collected on the 16 hybrid entries, the correlation between *S. frugiperda* injury ratings and predator profile data was not significant (*p* values > 0.05), except that the number of the hooded beetles was positively correlated to the 14 d and the mean of *S. frugiperda* injury ratings (*r* ≥ 0.18, *p* ≤ 0.03, *n* = 128) (Appendix A). The 7 and 14 d *S. frugiperda* injury ratings were positively correlated to the overall means of the *S. frugiperda* injury ratings, as shown in Appendix A.

Of all 11 taxa of predators recorded in the experiment in both years, seven predators were significantly correlated to the total number of predators (*r* values were between 0.20 and 0.9, *p* values ≤ 0.03, *n* = 128) (as shown in in last row of Appendix A). They were four lady beetles (the convergent, the pink spotted, the seven-spotted, and the multicolored Asian lady beetles), and two hemipteran predators (the big-eyed bug and the minute pirate bug), and the spiders.

### 3.4. Principal Component Analysis of S. frugiperda Resistance

The principal component analysis using the *S. frugiperda* injury ratings and predator data showed that Bt transgenic maize hybrids (CK1, CK3, and CK4) without *S. frugiperda* injury are located in quadrant III (Figure 2). Five reciprocal crosses had only moderate *S. frugiperda* resistance, which are in quadrants II and IV (Figure 2). However, because of relatively high *S. frugiperda* injury ratings, the reciprocal crosses between NC350 and NC358 (i.e., experimental hybrid entries 4A and 4B, as shown in Table 1) are in quadrant I with the susceptible check (CK2) (Figure 2). Similarly, the reciprocal crosses between NC358 and CML333 (or experimental hybrids 5A and 5B) are in quadrant II, and experimental hybrids 1A and 1B (reciprocal crosses between Oh43 and FAW1430) are in quadrant IV (Figure 2). Because each of the three reciprocal crosses (i.e., 1A and 1B, 4A and 4B, and 5A and 5B) are in the same quadrants (Figure 2), the principal component analysis confirmed these six experimental hybrids had no maternal/cytoplasmic effect, or the parents contributed similarly to *S. frugiperda* resistance. In contrast, the other three reciprocal crosses are different. Hybrid entries 2B, 3A, and 6A are in quadrant II, while the 2A, 3B, and 6B are in quadrant IV (Figure 2). Thus, the data for *S. frugiperda* injury ratings and numbers of predators among these reciprocal crosses were further examined to determine whether a maternal/cytoplasmic effect existed or not.

### 3.5. Maternal/Cytoplasmic Effect on Spodoptera frugiperda Resistance in Reciprocal Crosses

Based on the paired *t*-test results, as shown in Appendix A, the *S. frugiperda* injury ratings between each of the six reciprocal crosses were not different (*p* values ≥ 0.19), which means no maternal or paternal effect of the inbred lines on *S. frugiperda* resistance existed in any of these experimental hybrids. It is worth noting that there were no *S. frugiperda* injuries on Bt (DKC69-71), whereas its non-Bt version of its isogenic (sister) hybrid (DKC69-72) showed the highest injury ratings. In addition, the predator survey data recorded 7 d after the initial infestation were not different between each pair of reciprocal crosses (*p* values ≥ 0.09) (Appendix A).

## 4. Discussion

The *S. frugiperda* injury rating and predator data showed that the 12 experimental hybrids had relatively moderate resistance to *S. frugiperda* when compared to the three resistant checks (i.e., DKC69-71, DKC67-88 and P31P42 with transgenic Bt genes) and the susceptible check (non-Bt DKC69-72) of the experiment. Of the 12 experimental hybrids, two of them (i.e., FAW1430 × Oh43 and CML333 × NC358) showed lower *S. frugiperda* injury ratings than the rest of the 10 experimental hybrids (Figure 1).

Further comparisons between each pair of the six reciprocal crosses (or the 12 experimental hybrids) showed that, because no difference was detected between an inbred line as male or female parent in each set of the reciprocal crosses, male and female parents in the reciprocal cross contributed similarly to *S. frugiperda* resistance, or no maternal/cytoplasmic effect was observed. This is different from the previous report on maize weevil resistance, and maize weevil resistance in maize kernels is inherited maternally [12]. At the same time, the findings on *S. frugiperda* resistance from the current study are similar to the previous report that *H. zea* resistance was not related to maysin content in corn silk [13]. The previous study also reported that high levels of the maysin and its analogs that contributed to *H. zea* resistance did not show a maternal effect of parental inbred lines. The findings are valuable in developing maize germplasm with resistance to multiple biotic and abiotic stresses by targeting parental inbred lines with complementary traits of a specific parent (maternal or paternal) in designing the new crosses for both hybrid production and new germplasm development.

The nearly two-times higher number of predators recorded in 2014 when compared to 2013 suggests that the predation of *S. frugiperda* might have contributed to no difference in 14 d *S. frugiperda* injury ratings between the two years, given that the 7 day *S. frugiperda* injury ratings in 2013 were lower than in 2014. Such phenomena might not occur in northern states because of high latitudes with relatively low temperatures at the seedling stage of maize plants. In addition, although the current study confirmed previous reports that predator abundance during the experimental period is influenced greatly by weather conditions (e.g., temperature and rainfall) from year to year [6,20], further detailed studies are critical. Such studies would elucidate the tri-trophic interactions (i.e., maize genotype, *S. frugiperda* genotype, and predator abundance) modulated by the changing climate conditions and decipher the mechanisms or broad basis of host plant resistance to multiple biotic and abiotic factors. In particular, the use of four-row (or multiple-row) plots accompanied with a skip row spacing, instead of the continuous single-row plot as described in the current experiment, could be valuable in discerning dispersal/movement patterns, and the possible attraction of predators by a maize hybrid. The dataset from sampling the middle rows in multiple-row plots would represent predator abundance in a maize field with precision. In general, predator abundance in reducing foliar- and ear-feeding insects has been reported in warm climates in the recent literature. For example, both the pink spotted lady beetle and the insidious flower bug were abundant on maize plants at leaf stages 4–5 (V4 to V5), and natural enemies have the potential to be utilized for managing *S. frugiperda* in the subtropical lowlands of Mexico [35]. Earwigs were effective predators for the *S. frugiperda* larvae in Argentina [36]. The high survival rates of *Eriopis connexa* (Germar) (Coleoptera: Coccinellidae) fed with eggs of the *S. frugiperda* show the potential use of this prey in the laboratory rearing of lady beetles [37]. At the same time, earwigs, as the most abundant predators, should be further examined at the reproduction growth stage of maize plants for reducing kernel-destroying insect populations, e.g., *Helicoverpa zea* [38].

Regarding insect resistance mechanisms, the six pairs of reciprocal crosses were made by utilizing parents with relatively high levels of kauralexins and zealexins [18,19]. The two experimental hybrids showing the best *S. frugiperda* resistance were made by using three NAM founders (i.e., CML333, NC358, and Oh43), which have high levels of both phytoalexins (kauralexins and/or zealexins), and FAW1430, a newly developed inbred line from our program. Thus, the relationship of the phytoalexins (i.e., kauralexins, and zealexins) to foliar resistance to *S. frugiperda*, as well as the resistance mechanism of FAW1430 to *S. frugiperda* feeding, need to be further examined. In addition, plant-volatile (E)-β-caryophyllene, a terpenoid compound, is associated with *S. frugiperda* resistance in maize inbred line Mp708 [39]. The 4 d old *S. frugiperda* larvae preferred the whorl tissue Tx601 (susceptible control) over the Mp708 (resistant) tissue [39], which suggested non-preference (or antixenosis) of Mp708 as one of the possible resistance mechanisms. For multiple insect resistance mechanisms, by examining tropical maize inbred lines CML71, CML125, CML370, and CKSBL10008, with known resistance to multiple stem borers, revealed that CKSBL10008 and CML71 show high level of antibiosis to *S. frugiperda* [17]. In addition, by using extensive phenotyping data on *S. frugiperda* injury ratings from 289 maize germplasm lines, seven genes and multiple pathways associated with its feeding damage were identified [40]. This finding is critical to deciphering the molecular basis for further discerning the maternal/cytoplasmic effect of inbred parents on maize resistance to *S. frugiperda* and other key pests in hybrids, and subsequently to developing maize germplasm with resistance to multiple biotic and abiotic stress factors.

The current study also demonstrated that the three commercial maize hybrids with Bt transgenes were effective for controlling *S. frugiperda* (in quadrant III of Figure 2), while five of the six reciprocal crosses showed moderate resistance (in quadrants II and IV of Figure 2). The findings confirmed previous reports which showed that utilizing transgenic and native insect resistance could synergistically decelerate or even reverse Bt resistance in *S. frugiperda* and other lepidopteran pests in transgenic Bt maize production ([41,42], F.H., unpublished data).

In summary, the current study of unique reciprocal crosses determined that FAW1430 × Oh43 and CML333 × NC358 are the best performing hybrids. The two experimental hybrids will be further examined and utilized for developing new maize germplasm with resistance to multiple biotic and abiotic stresses. Comprehensive evaluation of *S. frugiperda* injury and predator abundance, as well as other ecological service factors at the seedling stage would be essential in understanding the ecological genetics of both plants and insect pests and their interactions with environmental factors. Given the current trend of increasing frequency and severity of abiotic stresses (e.g., heat and drought) in maize production, deciphering the genetic basis of dynamic plant responses to multiple biotic stress factors (e.g., insect infestations, phytopathogen infections, and subsequent mycotoxin accumulation) throughout the field season is essential to accelerating germplasm development, and ultimately minimizing losses of yield and quality in maize production.

## Figures and Tables

**Figure 1 insects-15-00289-f001:**
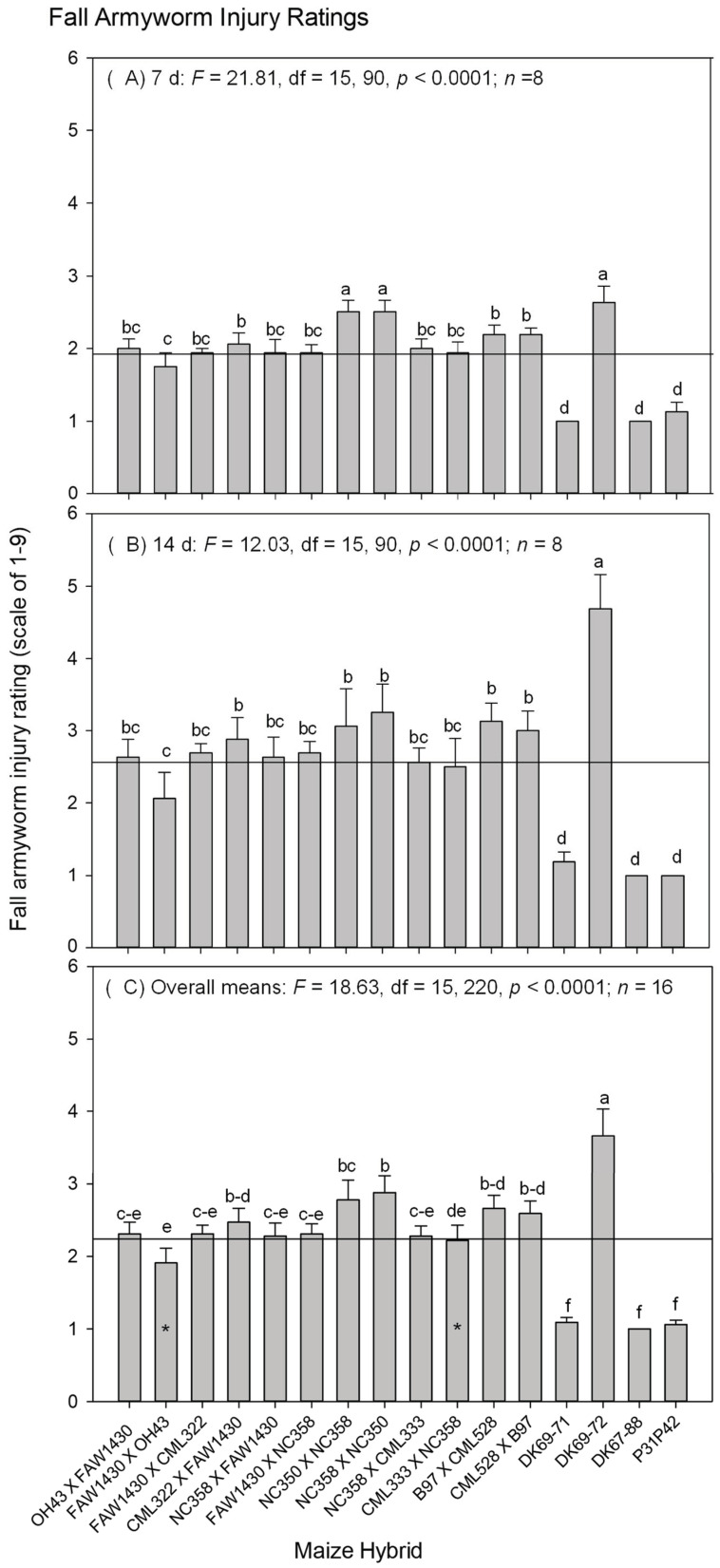
Combined two-year *S. frugiperda* injury rating data at the seedling stage: (**A**) 7 d rating data, (**B**) 14 d rating data, and (**C**) overall means from both ratings (7 and 14 d) of both years (2013 and 2014). The bars with * in (**C**) denote the experimental hybrids with the least *S. frugiperda* injury ratings.

**Figure 2 insects-15-00289-f002:**
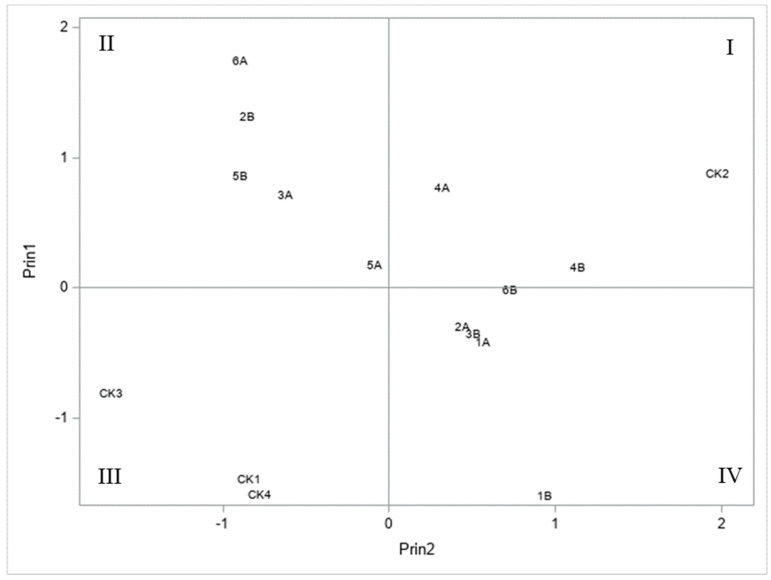
Identification of the best experimental maize hybrids from the six sets of reciprocal crosses and four commercial controls based on the principal component analysis results. The data on 14 d after *S. frugiperda* infestation, and predator survey data were used for the analysis. Principal component one (Prin1) contributed 52%, and principal component two (Prin2) contributed 48% of the overall variation in the experiment. The non-Bt commercial hybrid CK2 (DKC69-72) and 4A and 4B with the high *S. frugiperda* injury rating are in quadrant (**I**), whereas five hybrids (i.e., entries 2B, 3A, 5A and 5B, and 6A as shown in Table 1) are in quadrant (**II**). Three commercial hybrid checks (CKs) with transgenic Bt genes (i.e., CK1, CK3, and CK4 as shown in Table 1) are in quadrant (**III**) without *S. frugiperda* injury. Five hybrids (i.e., entries 1A and 1B, 2A, 3B, and 6B) are in quadrant (**IV**). The quadrants (**I**–**IV**) are labeled with the order used in mathematics.

**Table 1 insects-15-00289-t001:** The list of 12 experimental hybrids (or six pairs of reciprocal breeding crosses marked as A and B) and four commercial hybrids as the controls of the experiment in 2013 and 2014.

Entry	Pedigree (Female × Male)	References
1A	Oh43 × FAW1430	Oh43: [18,19,21]
1B	FAW1430 × Oh43	FAW1430: derived from FAW CC(C)5 [22]
2A	FAW1430 × CML322	CML322: [18,23,24]
2B	CML322 × FAW1430	
3A	NC358 × FAW1430	NC358: [18,19]
3B	FAW1430 × NC358	
4A	NC350 × NC358	NC350: [18,19]
4B	NC358 × NC350	
5A	NC358 × CML333	CML333: [18,19]
5B	CML333 × NC358	
6A	B97 × CML528	B97: [18,19]
6B	CML528 × B97	CML528: tropical inbred (procured from CIMMYT in 2011)
CK1	DKC69-71	Commercial check, Bt transgenic (Bayer, Monsanto)
CK2	DKC69-72	Commercial check, non-Bt control (Bayer, Monsanto)
CK3	DKC67-88	Commercial check, Bt transgenic (Bayer, Monsanto)
CK4	P31P42	Commercial check, Bt transgenic (DuPont, Pioneer)

## Data Availability

The data presented in this study are available within the article.

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
