# Peer review of "Field Evaluation of Experimental Maize Hybrids for Resistance to the Fall Armyworm (Lepidoptera: Noctuidae) in a Warm Temperate Climate"

_insects, 2024, doi:10.3390/insects15040289_

Round 1

Reviewer 1 Report

Comments and Suggestions for Authors

This manuscript develop maize germplasm with multiple pest resistance 15 and understand genetic inheritance, twelve experimental hybrids (six pairs of reciprocal crosses) 16 with diverse genetic backgrounds and four commercial checks were examined for FAW resistance 17 in 2013 and 2014. But there still have some point need revised.

1. All Figure are very poor. Figure 1 Y axis didn't have clearly unit and appropriate statistical methods. 

2. Figure 2 not clearly enough.

3. Method section should describe more details and clearly, including how many replications and the backgroud for all stains, how to survery and caculate the injury rating.

4. This manuscripts title is "Field Evaluation of Experimental Maize Hybrids for Resistance to the Fall Armyworm (Lepidoptera: Noctuidae) under Warm Temperate Climate" , but there didn' t have any climate temperate data in this survey.

5. The Table S1 some data are not normal.

6. The referances are not in the same format, need check again.

Comments on the Quality of English Language

Also the English language need some rewriting.

This manuscript require major corrections before it may be considered for publication.

Author Response

Reply to Reviewers’ Comments

Notes to Editors and Reviewers: The page and line numbers used in this reply is based on the attached copy named as “revised copy_insects-288412”, which is the copy with track of changes to show where the changes have been made.

            First of all, we greatly appreciate the two reviewers’ comments for strengthening our manuscript.  We have addressed all of reviewers’ concerns.  Our reply is listed as follows:

  1. Reply to Reviewer One’s Comments:
  2. All Figure are very poor. Figure 1 Y axis didn't have clearly unit and appropriate statistical methods. 

Reply: To address the reviewer’s concern for figure resolution, each of the original figures (generated using SigmaPlot) is included in a separate file named Figures.docx.  Each original figure is on a separate page. 

 Regarding y-axis label of Figure 1, the injury rating was made using a scale of 1-9 as described in the text, thus the label of y-axis is presented that way.  There was no unit for the rating, we just had 1-9 in the y-axis label to inform the range of the rating.

  1. Figure 2 not clearly enough.

Reply:  Figure 2 was copied from SAS printout, which is also in Figures.docx, as we mentioned in the reply above.  This file should have better resolution.  Please let us know if it is not.

  1. Method section should describe more details and clearly, including how many replications and the background for all strains?, how to survey and calculate the injury rating.

   Reply: We appreciate the comments.  These are important points, which should be described in the manuscript. We have checked the related sections, and found some of the information in the text the Reviewer asked.  Here they are: 1) the number of replications were described in three places: they are lines 18-19 (Abstract section), and lines 207 and line 218 (Sub-Sections of Study Site, Experimental Design and Data Analysis under the Materials and Methods section),

The strain of the fall armyworm and other related colony information was added, as shown in lines 158-161: “The S. frugiperda colony was established originally using insects collected from maize plants, feral insects were collected and added annually to the colony to maintain insect colony performance. The larvae of S. frugiperda used in this study was considered corn strain.”

Injury rating scale was described in Sub-Section 2.3, lines 174-186; predator survey method was described in Sub-Section 2.4, lines 188-200, and statistical analysis of injury rating and predate data was described in Subsection 2.5,  213-228.

  1. This manuscripts title is "Field Evaluation of Experimental Maize Hybrids for Resistance to the Fall Armyworm (Lepidoptera: Noctuidae) under Warm Temperate Climate" , but there didn' t have any climate temperate data in this survey.

   Reply: The reason we used manuscript title including “under a warm temperate climate” with two reasons;

  • The impressively abundant predators on maize seedlings are only observed under the warm climate in May to June annually; This may not be observed in northern (cool-temperate) climate, or high latitude.

  • Given the trend of climate change, high levels of insect and disease pressures, as well as the increasing levels of abiotic stress factors, the current study under the warm climate would be beneficial to future corn production in the Mid-West Corn Belt, and other regions worldwide in the northern climate or high latitude.

We have revised the following places to address these points in the first paragraph of Introduction (lines 38-62).

In addition, to explain the predator data in 2014 was nearly twice the number as what recorded in 2013, we have added a sentence for weather data collection in Materials and Methods section to understand weather influence on predator abundance:

  1. Lines 204-205: “Weather data used for the data analysis was obtained from Georgia Weather website (http://www.georgiaweather.net/).”
  2. Lines 225-228: “Weather data (i.e., daily high and low temperatures and rainfall) of May between 20213 and 2014 were compared (using the paired t-test) to understand environmental influence on frugiperda injury rating and predator abundance.”
  3. To make the weather data analysis meaningful in understanding predator abundance, the manuscript, the artificial fall armyworm infestation dates were added in Sub-Section 2.4, lines 188-190: “Both taxa and abundance of predatory arthropods from each experimental plot were recorded 7d after the initial frugiperda infestation (on May 19, 2013, and May 27, 2014, respectively) as described by previously [6, 20].”

Results section has also been revised to include weather data analysis results in relevance to predator abundance, which is shown in lines 310-320: “Comparison of weather data in May between 2013 and 2014 showed that daily high temperature was greater in 2014 (28.66 ± 0.65, n = 31) than in 2013 (26.66 ± 0.70, n = 31), and the difference was 2.01 ± 0.8 °C (n = 31, paired t-test t = 2.5, df = 30, P = 0.02), but neither daily low temperature nor rainfall was different (P values > 0.13).  Thus, low temperature in May of 2013 than that of 2014 might have led to less feeding by early instar larvae and subsequently low S. frugiperda injury ratings at day 7 in 2013 (Figure S1 A) when compared to the data of 2014 (Figure S2 A).  In contrast, nearly doubled number of predators recorded on day 7 after the artificial infestation in 2014 than that in 2013 (Table S1) caused by relatively high temperature might have contributed to the reduced S. frugiperda survival, due to predation, and subsequently no significant difference in the 14-d injury ratings between the two years.”

Discussion section has also been revised for understanding relevance of weather data-to both fall armyworm injury and predator abundance.  Please see lines 407-423 for details: “The nearly doubled number of predators recorded in 2014 compared to 2013 suggests that the predation of S. frugiperda might have contributed to no difference in 14-d S. frugiperda injury ratings between the two years.  Such phenomenon might not occur in northern states because of high latitude with relatively low temperature.  In addition, although the current study confirmed previous reports  that predator abundance during the experimental period is influenced greatly by the weather conditions (e.g., temperature and rainfall) from year to year [6, 20], further detailed studies are critical.  Such studies would elucidate the tri-trophic interactions (i.e., maize genotype, S. frugiperda genotype, predator abundance) modulated by the changing climate conditions, and decipher the mechanisms or broad basis of host plant resistance to multiple biotic and abiotic factors.  In particular, the use of four-row (or multiple-row) plots accompanied with a skip row spacing, instead of the continuous single-row plot as described in the current experiment, could be valuable in discerning dispersal/movement patterns, and possibly attraction of predators by a maize hybrid.  The dataset from sampling middle rows in multiple-row plots would represent predator abundance in a maize field.”

  1. The Table S1 some data are not normal.

   Reply:  The table was a direct printout of statistical software (SAS), and the data were checked again.  Because the table format was not correct in the manuscript, we have reformatted both tables S1 and S2.  Please see the tables in excel format. We hope this would be easy to follow the data as shown (see attached file named Table S1.xlsx). 

In addition, Table S2 has also been reformatted as well to present the data clearly by adding a spacing row between each parameter (see Table S2.xlsx for details).

  1. The references are not in the same format, need check again.

   Reply:  THANK YOU! It is awfully embarrassing to admit this, but I do.  I do greatly appreciate the reviewer’s comment on this. The reference list has been thoroughly checked more than once, and matched, and reformatted by following the journal (Insects) guidelines.

Reviewer 2 Report

Comments and Suggestions for Authors

All specific concerns, edits for improvement are listed in the margin of the enclosed file.

Overall, this is a useful, straightforward analysis of native and Bt based hybrids for HPR against FAW. My only concern with the paper is the detailed emphasis and discussion of the predator data. In brief, because "single-row plots" were used (and no mention of a skip row; or distance between rows)...it is not at all surprising that there are No to few sig. diffs. (P>0.05) among hybrids. Predator movement is not discussed, yet it is well known that most of the spp mentioned have dispersal capacity to move within the season from row to row, (plot to plot). To truly assess the role of hybrids on predators, much larger plot sizes would be needed; e.g, at least 8-row plots, and data only taking in the middle 2-rows, etc.  The lack of NS diffs in the system was cited by some of the same authors' previous studies (were those also single-row hybrids). I understand the interest in including predator effects as this could in fact reduce FAW injury levels. However, the authors should consider reducing the text on predators overall; e.g., talking about how the most dominant lady beetle spp. are correlated with Total predator numbers -- without mention of biological relevance for this, etc. -- is not useful. Could delete this text.

The main point is that Total predators in 2014 "may have " had an impact on FAW injury levels.  State this and move on to the main focus of this paper, which is Efficacy of X native resistance, and similarities to Bt events.

Not sure if the authors mention current evidence or concern of FAW resistance to Bt --as a major issue--but should be in the Disc.

There's also no mention of the numerous parasitoid spp. on FAW that should at least be mentioned in the Disc; sample text included.

Comments on the Quality of English Language

Several minor edits were made; not all paragraphs checked

Author Response

Reply to Reviewers’ Comments

Notes to Editors and Reviewers: The page and line numbers used in this reply is based on the attached copy named as “revised copy_insects-288412”, which is the copy with track of changes to show where the changes have been made.

            First of all, we greatly appreciate the two reviewers’ comments for strengthening our manuscript.  We have addressed all of reviewers’ concerns.  Our reply is listed as follows:

 A. Reply to Reviewer Two’s Comments in Review Report :

  1. All specific concerns, edits for improvement are listed in the margin of the enclosed file.

Reply: We appreciate the Reviewer’s constructive comments, as well as his/her thoroughness for strengthen the manuscript.  We have made all corrections throughout the text.

  1. Overall, this is a useful, straightforward analysis of native and Bt based hybrids for HPR against FAW. My only concern with the paper is the detailed emphasis and discussion of the predator data. In brief, because "single-row plots" were used (and no mention of a skip row; or distance between rows)...it is not at all surprising that there are No to few sig. diffs. (P>0.05) among hybrids. Predator movement is not discussed, yet it is well known that most of the spp mentioned have dispersal capacity to move within the season from row to row, (plot to plot). To truly assess the role of hybrids on predators, much larger plot sizes would be needed; e.g, at least 8-row plots, and data only taking in the middle 2-rows, etc.  The lack of NS diffs in the system was cited by some of the same authors' previous studies (were those also single-row hybrids). I understand the interest in including predator effects as this could in fact reduce FAW injury levels. However, the authors should consider reducing the text on predators overall; e.g., talking about how the most dominant lady beetle spp. are correlated with Total predator numbers -- without mention of biological relevance for this, etc. -- is not useful. Could delete this text.

Reply: We appreciate the Reviewer’s comments above, as well as on the scanned copy of the manuscript.  The comments were on page 4, line 146 about experimental design, and page 5 line 180 about predator survey. We have addressed the reviewer’s concern thoroughly. Our reply is as follows:

  • For comments at the end of Sub-Section 2.1 about experimental design, we revised two places in the text to address the two concerns:

  1. a) For the use of single-row plot, we revised page 5, Sub-Section 2.5, lines 209 by adding “Due to the limited seed supplies of the 12 selected experimental hybrids, which were from a diallel cross plan, a single-row plot was used in the current study.”
  2. b) Sub-Section 3.3 of Results was significantly reduced based on the Reviewer’s comments for the section, please see lines 328-333 for the changes: “Of all 11 predators recorded in the experiment in both years, seven predators were significantly correlated to the total number of predators (r values were between 0.20 and 0.9, P values ≤ 0.03, n = 128) (as shown in in last row of Table S2). They were four lady beetles (the convergent, the pink spotted, the seven-spotted, and the multicolored Asian lady beetles, and two hemipteran predators (the big-eyed bug and the minute pirate bug), and spiders.”
  • For future study of improving the precision of predator survey, we extensively revised the following section in Discussion (lines 407-423) to address the improvement in predator survey: “The nearly doubled number of predators recorded in 2014 compared to 2013 suggests that the predation of frugiperda might have contributed to no difference in 14-d S. frugiperda injury ratings between the two years.  Such phenomenon might not occur in northern states because of high latitude with relatively low temperature.  In addition, although the current study confirmed previous reports  that predator abundance during the experimental period is influenced greatly by the weather conditions (e.g., temperature and rainfall) from year to year [6, 20], further detailed studies are critical.  Such studies would elucidate the tri-trophic interactions (i.e., maize genotype, S. frugiperda genotype, predator abundance) modulated by the changing climate conditions, and decipher the mechanisms or broad basis of host plant resistance to multiple biotic and abiotic factors.  In particular, the use of four-row (or multiple-row) plots accompanied with a skip row spacing, instead of the continuous single-row plot as described in the current experiment, could be valuable in discerning dispersal/movement patterns, and possibly attraction of predators by a maize hybrid.  The dataset from sampling middle rows in multiple-row plots would represent predator abundance in a maize field.”

3.The main point is that Total predators in 2014 "may have " had an impact on FAW injury levels.  State this and move on to the main focus of this paper, which is Efficacy of X native resistance, and similarities to Bt events.

Reply: We appreciate this point for strengthening the manuscript. We have addressed the reviewer’s comments #3 ad #4 together, and revised Discussion section by revising lines 469-475: “The current study also demonstrated that the three commercial maize hybrids with Bt transgenes were effective for controlling S. frugiperda (quadrant III in Figure 2), while five of the six reciprocal crosses showed moderate resistance (quadrants II and IV in Figure 2).  The findings from the present experiment confirmed the previous reports that utilizing transgenic and native insect resistance in complementary could synergistically decelerate or even reverse Bt resistance in S. frugiperda and other lepidopteran pests in transgenic Bt maize production[41, 42, 43].”

Three additional references are added for the revised discussion:

Ni, X.; Lei, Z.; He, K.; Li, X.; Li, X.; Xu, W. Integrated pest management is the lucrative bridge connecting the ever-emerging knowledge islands of genetics and ecology. Insect Science, 2014b, 21(5), 537-540.

Huang, K.; He, H.; Wang, S.; Zhang, M.; Chen, X.; Deng, Z.; Ni, X.; Li, X. Sequential and simultaneous interactions of plant allelochemical flavone, Bt toxin Vip3A and insecticide emamectin benzoate in Spodoptera frugiperda. Insects, 2023, 14, article736. https://doi.org/10.3390/insects14090736.

Silva, T.; Sword, G.A.; Miller, A.; Qureshi, J.A.; Head, G.P.; Kerns, D.D.; Jurat-Fuentes,J.L.L.; Villegas, J.; Towels, T.; Ni, X.; Reay-Jones, F.P.F.; Carrillo, D.; Cook, D.R.; Daves, C.; Stout, M.J.; Thrash, B.; Paula-Moraes, S.; Lin, S.; Patla, B.; Niu, Y.; Sakuno, C.R.I.; Huang, F. Reversibility of practical resistance to Bt crops: a case report on the resistance of fall armyworm to Cry1F maize in the southeastern United States. Journal of Pest Science. (in review)

4.Not sure if the authors mention current evidence or concern of FAW resistance to Bt --as a major issue--but should be in the Disc.

Reply: We appreciate this comment.  As we mentioned previously, we revised Discussion section by adding lines 469-475 (as shown above for comment #3) to address this point. 

5.There's also no mention of the numerous parasitoid spp. on FAW that should at least be mentioned in the Disc; sample text included.

Reply: We have addressed the reviewer’s concern, and revised Materials and Methods section to address this point in Sub-Section 2.4, in lines 197-200.  We added the following sentences:

“Although parasitoids are often associated S. frugiperda infestations in maize fields [31, 32, 33], few parasitoids were observed 7 d after the artificial infestation with neonate larvae of S. frugiperda in the spring.  Thus, the parasitoids were not included in the sampling.”

Three references on parasitoids of fall armyworm are added:

Meagher, R.L; Nuessly, G.S.; Nagoshi, R. N.; Hay-Roe, M.M. (2016) Parasitoids attacking fall armyworm (Lepidoptera: Noctuidae) in sweet corn habitats. Biological Control, 95, 66-72. https://doi.org/10.1016/j.biocontrol.2016.01.006.

Abbas, A.; Ullah, F.; Hafeez, M.; Han, X.; Dara, M.Z.N.; Gul, H.; Zhao, C.R. Biological control of fall armyworm, Spodoptera frugiperda. Agronomy2022. 12(11), 2704.

Pal, S.; Bhattacharya, S.; Dhar, T.; Gupta, A.; Ghosh, A.; Debnath, S.; Gangavarapu, N.; Pati, P.; Chaudhuri, N.; Chatterjee, H.; Senapati, S.K.; Bhattacharya, P.M.; Gathala, M.K.; Laing, A.M. (2024) Hymenopteran parasitoid complex and fall armyworm: a case study in eastern India. Scientific Reports 14, 4029. https://doi.org/10.1038/s41598-024-54342-z

B. Reply to Reviewer Two’s Remarks and Advice on the Scanned Copy

First of all, we greatly appreciate Reviewer Two’s thorough corrections and constructive comments on the scanned copy of the manuscript for strengthening the manuscript.  Our specific replies to the comments are listed as follows:

  • All grammatical errors were corrected throughput the text.
  • For critical comments for Sub-Section 3.3 on pages 8-9, we have shortened the section, as shown in our reply for Reviewer Two, comment #2, that is, or lines 328-333.

We also separated the sub-section 3.3 into two paragraphs. While the first paragraph discussed fall armyworm injury and predator correlation, the second paragraph discussed the correlation among the 11 predators.   

Also, we thought presentation of Table S2 that shows correlation among fall armyworm injury and 11 predators recorded, is important to understand tri-trophic interactions in relation to host plant resistance modulated by the changing climate.  Please see lines 407-423 in Discussion section for details.

  • Page 10, comments next to line 373: please see above reply for comment #5 from Reviewer Two. We addressed this concern in lines 197-200.
